# Assessing the Economy for the Common Good Measurement Theory Ability to Integrate the SDGs into MSMEs

**Ana T. Ejarque and Vanessa Campos *** 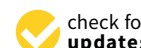

Business Administration Department, Faculty of Economics, University of València, 46022 València, Spain; Ana.T.Ejarque@uv.es

* Correspondence: Vanessa.Campos@uv.es

**Abstract:** Over the past decades, sustainability and corporate sustainability have gained a lot of attention. Currently, the focus of attention has shifted to the integration of the Sustainable Development Goals (SDGs) into businesses operation. The extant literature points to the proposed frameworks as not fitting micro, small, and medium-sized enterprises (MSME) reality and, also, to a lack of empirical evidence in this field. With research at the intersection of business and SDGs still being scarce, the Economy for the Common Good (ECG) model allows operationalizing the SDGs employing its novel measurement theory. The present study is aimed at completing the statistical validation process of the ECG measurement theory using confirmatory factor analysis (CFA) on a sample of 206 European firms. Thus, after having performed an exploratory factor analysis (EFA), this study takes as a starting point the previously published knowledge and proceeds with the second step of the statistical validation process. The results of CFA confirm the conclusions of the EFA and allow to redefine the measurement scales included in the ECG framework to achieve a sufficient level of goodness of fit.

**Keywords:** corporate sustainability; sustainability management tools; economy for the common good; sustainable development goals; confirmatory factor analysis

---

## 1. Introduction

Over the last two decades, business environments have rapidly evolved towards corporate sustainability [1]. As a result, companies are more aware of improving economic, environmental, and social performance simultaneously [2].

Similarly, several authors point out the huge increase of indicators and methods to measure sustainable development [3], as well as a new non-financial reporting framework from a social and environmental point of view, thus giving birth to integrated reporting.

The United Nations defined the Sustainable Development Goals (SDGs) in 2015 as an international guideline to achieve human wellbeing and environmental preservation, understood as social inclusion, respect for everyone, and human dignity [4]. Thus, both organizations and countries have adopted different sustainable indicators to manage and monitor sustainable development-related matters (Allen et al., 2017). In this context, the next step for sustainability management and control tools is to allow the integration of the SDGs into strategic management since these types of decisions are made at a strategic level [1]. However, these tools are not usually adapted to be applied to small or medium-sized enterprises (SMEs). In other cases, the difficulty appears when translating and adapting them into a specific industry or legislation [5].

Thus, the Economy for the Common Good (ECG) model by Felber [6,7] arises as an alternative sustainability management and control framework which is being implemented in several European

businesses, mainly in German-speaking countries. The ECG model, as a sustainability management and control system, works utilizing two interconnected tools the Common Good Matrix (CGM) and the Common Good Balance Sheet (CGBS) [8].

In this sense, Engert et al. [1] performed an exhaustive literature review on the topic and concluded that there is a need to foster empirical research in this field, i.e., the integration of corporate sustainability into business management. This paper is aimed at analyzing the measurement theory proposed by the ECG model, thus, assessing its statistical validity and reliability. To do so, we employed confirmatory factor analysis (CFA) given that we have already conducted exploratory factor analysis (EFA) [8]. Therefore, the present work is the following step in the EGG measurement theory validation process.

This paper is structured as follows. Section 2 presents the theoretical framework involving an overview of corporate sustainability (CS), integrated reporting (IR), SDGs, and how the ECG model allows the operationalization of these concepts in the business context. Section 3 describes the research process and the methodology employed. Then, Section 4 presents the main findings. Finally, Section 5 depicts the discussion and conclusions.

## 2. Theoretical Framework

### 2.1. Corporate Sustainability and Integrated Reporting

The concept of corporate sustainability (CS) has its origins in the relationship between corporate social responsibility (CSR) and sustainability. The Brundtland Commission defined sustainable development as one which meets the needs of the present without compromising the ability of future generations to meet their own needs [9]. Bansal points out three main sustainable principles: environmental integrity (guarantees that human activities do not compromise natural resources and biodiversity), economic prosperity (which implies that distribution and creation of goods and services help raise the standard of living throughout the world), and social equity (guarantees that all members of society have equal access to opportunities and resources) [10]. In other words, CS is about making compatible economic viability, whole respect for the environment, and being socially equitable and ethical [11].

In the last twenty years, some scholars have provided different definitions of CS, on the assumption that this subject is the business approach that deals with sustainable development. Thus, Bos-Brouwers [2] noted that CS is aimed at improving the economic, environmental, and social performance of companies, and is also recognized as the triple P of business, namely: people, planet, and profit. In the same way, Lozano [12] defined CS as the corporate activities that proactively attempt to contribute to sustainability equilibrium, including the economic, environmental, and social dimensions of today, as well as their inter-relations within and over the time dimension while addressing the company's systems, as well as its relationship with its stakeholders. Jung and Jung [13] provide the third definition of CS as the consecution of economic, social, and environmental goals through a legal business entity meeting the needs of the present without compromising the ability and capacity of future generations to meet their own needs. In this sense, all of these definitions of CS point to the need to integrate and combine economic, social, and environmental aspects in firms' management [11].

In light of this, several authors agree that CS is achieved at the intersection of economic development, environmental protection, and social responsibility. This entails considering a holistic perspective, understood as the need to consider all three dimensions (economic, environmental, and social). Such a vision is also reflected in the concept of the "triple bottom line" [14], as well as their impacts.

By its side, the ISO 8420 defined total quality management (TQM) as a management approach focused on quality, taking into account the participation of all its members with a long-term success goal, oriented not only to customer satisfaction but also to benefits for all members (of the organization and for society) [15]. Thus, this definition would be strongly connected to the stakeholder approach [16–18].

Under those circumstances, CS requires managers to address interconnected concerns for the natural environment, social welfare, and economic prosperity at one time [19]. Corporate sustainability management is defined as a response to environmental and social issues arising from the organization's primary and secondary activities, in strategic and profit-driven corporate terms [20]. Therefore, organizations have to implement concepts and systems, as well as management instruments, i.e., sustainability management tools, to operationalize social and environmental sustainability. In other words, managers have to consider different aspects of CS and integrate them into their corporate strategy, making sure that effectiveness is being considered and long-term goals can be accomplished [1].

In this line, Porter and Kramer [21] suggested shared value creation as the starting point to redefine capitalism by creating economic value and social value simultaneously, while addressing its needs and challenges. Thus, a company should plan its business based on society and its problems, rather than the business itself, to open business opportunities in society [22]. However, Crane et al. [23] pointed out that shared value creation is focused on those monetary issues and concerns by promising economic value for businesses, therefore it is unlikely to be a sufficient approach for solving social problems. In the same way, Dyllick and Hockerts [11] found that businesses should go beyond eco-efficiency and socio-efficiency in a time that addresses the real sustainability issues their societies are facing.

With this in mind, one can realize how, in terms of social purpose, there is a need for new organizational forms. Thus, Dylick and Muff [22] point out social business, social entrepreneurship, B-corporations, and the ECG model as alternative organizational models. These authors distinguished between four sustainability approaches based on inputs, the values created, and the organizational processes involved: (a) the current paradigm, understood as a purely economic view focused on profits, market value, and shareholder value; (b) shareholder value-oriented, namely introducing social and environmental concerns into the current paradigm without variating the main business outlook, for the purpose of reducing costs and increasing reputation, profits, competitiveness, market positions, and shareholder value; (c) the triple bottom line approach, perceived as a further step beyond shareholder value, by integrating social and environmental issues into the planning business and reporting on measurable results about the achievements in an externally transparent form; and (d) common good value-oriented, from exploring how to minimize negative impacts to understanding how the company can create a positive impact on society and the planet as a whole, by contributing to transparency, sharing best practices, and establishing common actions and standards.

Therefore, CS means achieving long-term economic success while combining issues overcoming disputes of purposes between economic, environmental, and social issues. To do so, CS needs to become part of the company's strategy (vision, culture, governance, performance, and management simultaneously).

Besides, one can appreciate how in terms of organizational performance, there exists an increasing concern on the creation of value for people, society, and the environment. As a consequence, the traditional financial business reporting model needs to evolve towards corporate sustainability management and control (reporting) tools. Thus, it is possible to demonstrate results by measuring progress and clarifying consistency between activities, outputs, outcomes, and goals [24]. According to Waddock [25], stakeholders are demanding significantly more revelations related to a corporation's environmental and social practices, apart from economic performance. In other words, non-financial measurements need to be reflected and included in the integration of CS into strategic management [1].

Hence, Dumay et al. [26] conclude that traditional corporate reporting does not appropriately satisfy the information needs of stakeholders in evaluating an organization's performance. Under those circumstances, scholars and practitioners gave birth to the field of IR by developing a new non-financial reporting framework from a social and environmental point of view.

In the present times, the Global Reporting Initiative (GRI) has led to the most extended non-financial reporting framework. The Coalition for Environmentally Responsible Economies (CERES) founded the GRI in 1997 to create a globally applicable sustainability reporting framework [27]. Since then,

its following versions have been updated with a stronger emphasis on clarity, the purpose of criteria, and the process of reporting. Up to July 2018, the operative version was G4 built up in 2013 and launched in 2014. Nevertheless, from July 2018, a new version that interrelates four modules (universal, economic, environmental, and social) has substituted G4. Additionally, its sustainability reporting guidelines were recognized in the World Summit on Sustainable Development Plan of Implementation. For this reason, the GRI is displayed in a range of influential and inter-connected international institutional settings [28].

In 2010, the International Integrated Reporting Council (IIRC), formed by a global coalition of regulators, companies, investors, standard setters, accountants, and non-governmental organizations (NGOs), developed a global integrated report (IR) for the first time to develop a set of internationally accepted corporate reporting rules and to overcome the existing problems of over-information, lack of clarity, and reliability [29].

As reported by IIRC (http://integratedreporting.org), "an IR is a concise communication about how an organization's strategy, governance, performance, and prospects, in the context of its external environment, lead to the creation of value in the short, medium and long-term". Namely, IR comprises the crucial financial, social, environmental, and corporate governance information by compressing it in one report. Therefore, IR is seen as the natural next step as it goes beyond sustainability reporting [28]. Thus, an IR must include: (1) a general vision of the organization and its environment (the political, legal, social, and environmental issues that can affect the organization and its value creation as well as its scope); (2) governance (focused on how the organization's governance structure is and how it supports its ability to create value in the short, medium and long term); (3) business model (how the organization creates value); (4) risk and opportunities (specify the main risks and opportunities affecting the organization and how they can deal with them to create value); (5) strategy and resource allocation (what is the organization's ultimate purpose and how it will achieve it); (6) performance (strategic goals within the timescale); (7) outlook (defines the organization's main challenges and uncertainties to implement its strategy); and (8) basis of preparation and presentation (determination of the relevant aspects to be integrated into the report and how they are quantified and evaluated).

Equally important is the European Directive 2014/95/UE which set up the duty of producing non-financial statements (NFSs) for large firms. Such NFSs must incorporate information related to (1) a brief business model description (activities performed and indispensable information about how these activities are accomplished), (2) a clarification on policies and procedures (related to human rights, environmental and social concerns, staff, and corruption prevention), (3) how the issues included in point 2 can be associated with the firm's core businesses and its main risks, and (4) key non-financial indicators (KPI), relevant to the firm's core business. In case these indicators were not provided, firms should indicate the reason(s) why they were not disclosed.

Thus, the ECG model relies on two tools to operationalize and integrate CS into the business context, i.e., the CGM and the CGBS. The CGM is the tool that guides the implementation process. It is conceived as a strategic matrix to guide the integration of sustainability strategies into the business operation. To do so, the CGM takes stakeholders' management as a reference and drives it according to four cross-values: human dignity, solidarity and social justice, environmental sustainability, and transparency and co-determination. Associated with the CGM, the ECG model proposes a set of indicators to monitor the process evolution which constitutes the ECG measurement theory. By its side, the CGBS takes such a set of indicators as a starting point and works as an integrated report that allows the process monitoring. The main novelty of the CGBS as an integrated report, however, is that it works as a source of information related to sustainability concerns for both internal and external stakeholders [8].

Finally, it is worth mentioning that Ketola [30] has also proposed the idea of employing a strategic matrix to support the implementation of CS in the business context, i.e., the corporate responsibility portfolio matrix. However, such a matrix did not work together with any type of integrated report.

Figure 1 below shows the CGM version 5.0. Its rows depict the five groups of stakeholders and its columns specify the cross-values that drive the stakeholders' management. To measure the degree of accomplishment, every one of its cells proposes indicators, thereby constituting a measurement theory according to the definition by Hair et al. [31].

| VALUE / STAKEHOLDER | HUMAN DIGNITY | SOLIDARITY AND SOCIAL JUSTICE | ENVIRONMENTAL SUSTAINABILITY | TRANSPARENCY AND CO-DETERMINATION |
|---|---|---|---|---|
| **A: SUPPLIERS** | **A1** Human dignity in the supply chain | **A2** Solidarity and social justice in the supply chain | **A3** Environmental sustainability in the supply chain | **A4** Transparency and co-determination in the supply chain |
| **B: OWNERS, EQUITY- AND FINANCIAL SERVICE PROVIDERS** | **B1** Ethical position in relation to financial resources | **B2** Social position in relation to financial resources | **B3** Use of funds in relation to the environment | **B4** Ownership and co-determination |
| **C: EMPLOYEES** | **C1** Human dignity in the workplace and working environment | **C2** Self-determined working arrangements | **C3** Environmentally friendly behaviour of staff | **C4** Co-determination and transparency within the organisation |
| **D: CUSTOMERS AND BUSINESS PARTNERS** | **D1** Ethical customer relations | **D2** Cooperation and solidarity with other companies | **D3** Impact on the environment of the use and disposal of products and services | **D4** Customer participation and product transparency |
| **E: SOCIAL ENVIRONMENT** | **E1** Purpose of products and services and their effects on society | **E2** Contribution to the community | **E3** Reduction of environmental impact | **E4** Social co-determination and transparency |

**Figure 1.** The Common Good Matrix 5.0.

## 2.2. Sustainable Development Goals and Economy for the Common Good

In the present times, several organizations have adopted sustainable development indicators and composite indicators to report and monitor their advances concerning sustainable development. Thus, the novel adoption of the Sustainable Development Goals (SDGs) confirms their increasing importance in terms of decision making [3].

The United Nations defined 17 SDGs to track the economic, social, and environmental challenges, by offering specific targets (169 in total) and indicators (230 in total). Thus, the 17 goals can be classified into five themes: people, planet, prosperity, peace, and partnership. As a result, the United Nations provides an overview of the 17 SDGs: (1) end poverty in all its forms everywhere; (2) end hunger, achieve food security and improved nutrition, and promote sustainable agriculture; (3) ensure healthy lives and promote well-being for all at all ages; (4) ensure inclusive and equitable quality education and promote lifelong learning opportunities for all; (5) achieve gender equality and empower women and girls; (6) ensure availability and sustainable management of water and sanitation for all; (7) ensure access to affordable, reliable, sustainable, and modern energy for all; (8) promote sustained, inclusive, and sustainable economic growth, and full and productive employment and decent work for all; (9) build resilient infrastructure, promote inclusive and sustainable industrialization, and foster innovation; (10) reduce inequality within and among countries; (11) make cities and human settlements inclusive, safe, resilient, and sustainable; (12) ensure sustainable consumption and production patterns; (13) take urgent action to combat climate change and its impacts; (14) conserve and sustainably use the oceans, seas, and marine resources for sustainable development; (15) protect, restore, and promote sustainable use of terrestrial ecosystems, sustainably manage forests, combat desertification, halt and reverse land degradation, and halt biodiversity loss; (16) promote peaceful and inclusive societies for sustainable development, provide access to justice for all, and build effective, accountable and inclusive institutions

at all levels; and (17) strengthen the means of implementation and revitalize the global partnership for sustainable development [32].

In contrast to the Millenium Development Goals (MDGs), which expired in 2015, the SDGs have a wider scope. Consequently, different from the MDGs' approach focused on human development through poverty alleviation, the SDGs provide a more holistic scope by capturing aspects from the triple bottom line (more economic, social, and environmental-related concerns) closer to the sustainability approach. Moreover, SDGs propose an increasing concern related to intangible aspects like inclusion, dignity, and justice to be applied to all countries [33].

In this context, the SDGs aim at driving and enhancing the engagement of stakeholders. Hence, the United Nations developed them by adopting a multi-stakeholder approach, which includes national, sub-national, and local governments, academia, civil society organizations, development partners, and businesses, since the SDGs differentiate between national and local stakeholders-levels [5].

According to Verboven and Vanherck [5], the SDGs were designed to be applicable at the national level, and in both developing and developed countries. However, given the difficulties in monitoring all of the 230 indicators proposed, each country should select specific indicators that fit with national development priorities and strategies [3].

Moreover, the United Nations developed the SDG Compass, a guideline aimed at advising companies on how to align their strategies while measuring and managing their contribution to the SDGs. However, Verboven and Vanherck [5] hold that the SDG Compass is addressed to multinationals and large companies, whilst another key point is the need to also apply the SDGs to micro-, small-, and medium-sized enterprises (MSMEs). To do so, MSMEs need to integrate the SDGs into their strategies and operationalize them through management tools. Thus, sustainability should be integrated into the organization's strategy and daily business operations, enabling material outcomes [34].

In the European MSMEs context, some of the SDG targets are difficult to translate and adapt because they are out of scope or are the subject of legislation, e.g., targets concerning minimum wage and gender parity. For this reason, adjusting the SDGs' targets is very challenging and time-consuming for European MSMEs. In other words, it requires the development of specific sustainability management tools.

In terms of developing an effective sustainability tool, usability and applicability are fundamental features. In this sense, Verboven and Vanherck [5] reported that an operative sustainability tool needs a holistic method which allows a wider sustainability approach as well as create an impact at the strategic, tactical, and operational level [33]. Likewise, the sustainability management and control tool should provide a detailed vision of topics by offering an effective translation of the topics into indicators. Therefore, the framework should distinguish between the management process and the thematic framework and also facilitate an analytical part that generated a report. In summary, the framework is required to be flexible and user-friendly in every business context.

According to the above-mentioned, the adoption of sustainability strategies at the organizational level through the SDGs requires the integration of sustainability management and reporting into a single framework. Given that, we argue that the ECG model provides a framework to do it. Thus, the CGM and the CGBS facilitate the operationalization of SDGs' sustainability management and reporting [35,36]. More recently, some authors [37] have associated the different cells and indicators of the CGM to the SDGs holding that the ECG model is an effective framework to integrate the SDGs into the business operation, hence providing theoretical evidence of face validity concerning the ECG measurement theory and its ability to integrate the SDGs into business management. However, they did not provide empirical evidence to support their arguments. Consequently, this paper tries to fill this gap by providing empirical evidence based on a sample of 206 European businesses.

To summarize, we argue that the CGM and the CGBS are tools that can facilitate the management and monitoring of firms' behavior in terms of social and environmental concerns. Furthermore, the ECG model allows its implementation by any type of organization, including MSMEs, as the model provides a simplified version specifically designed for MSMEs. This way, the ECG framework provides

an answer to social and environmental needs by developing new stakeholder relations and reinforces economic value creation simultaneously, therefore levering social and entrepreneurial innovation processes [38].

Finally, the present work is aimed at assessing the statistical validity of the ECG measurement theory to provide an answer to our research question: "Are the measurement scales of the CGM valid and reliable from a statistical point of view?" For that reason, we transformed the constructs and items proposed by the ECG measurement theory into a research model. Figure 2 below depicts our research model.

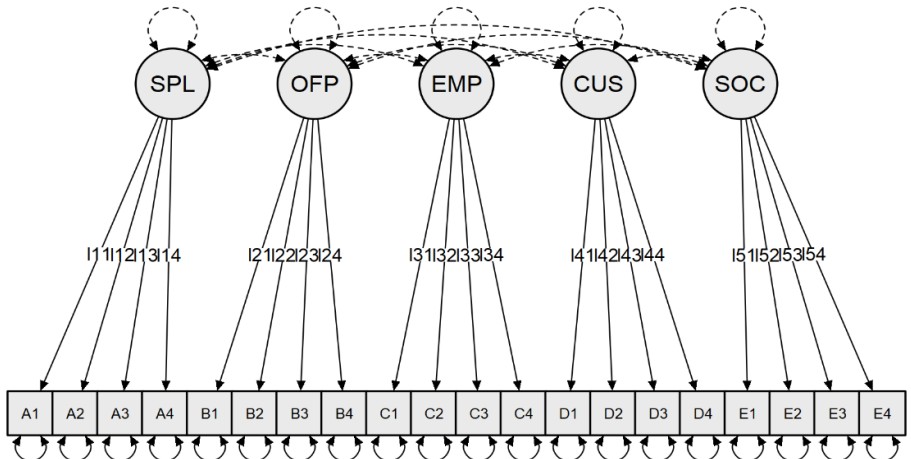

**Figure 2.** Research model: 5 factors and 20 items.

## 3. Methodology

To test the ECG model's measurement theory (operationalized employing the CGM and the CGBS), we designed a cross-sectional study based on a questionnaire distributed among the firms that have implemented the ECG model from 2011 to 2017 in Europe. The questionnaire asked the firms about the scores they have obtained in the different items included in the CGM and reported in the CGBS. It also picked up information on the industry, age, country of origin, number of employees, and turnover, these variables being treated as control variables for statistical purposes.

Thereafter, we distributed the questionnaire through an e-mail addressed to the firms' managers during the first quarter of 2018. The e-mail contained a link that allowed the firms to fulfill the questionnaire on the online platform SurveyMonkey; they could also upload their CGBS to the platform or send it by e-mail. This facilitated the data-gathering as it enabled us to download the data matrix directly from the online platform, then we only had to type the scores of the firms that had opted for uploading their CGBS or sending them by e-mail.

The population overall comprised of 400 European firms that had implemented the ECG model by producing and auditing their CGBS up to December 31, 2017. We sent the questionnaire to the overall population and got 206 full and valid responses, i.e., the sample comprised 51.50% of the population.

Five European countries concentrate most of the population of firms working under the ECG framework: Germany (39.8%), Austria (30.1%), Spain (19.4%), Italy (7.8%), and Switzerland (2.4%). The rest of the European countries account for 0.49% of the population.

When applying the ECG framework, the firms can obtain a maximum score of 1000 points by applying the measurement scales included in the CGM and reported in the CGBS. The average score obtained by the firms included in the sample was 497, the median was 498. Thus, according to the rating employed by the CGBS [7], most of the firms fell in the "experienced" level (CGBS score between 301 and 600 points). Specifically, 67.96% of firms in the sample fell in the "experienced" level, 24.27% of them fell in the "exemplary" level (between 601 and 1000 points). None of them fell into the "beginner"

level (between 1 and 100 points) and only 7.77% of them fell into the "advanced" level (between 101 and 300 points).

As the last purpose of the current study is to statistically test and validate the ECG model's measurement theory, in our research model we defined the dimensions (constructs/factors) and items in the way they are designed and associated in the 5.0 version of the CGM and the CGBS (the version currently in force).

Furthermore, given that the present study includes the European firms that have implemented the ECG model producing their CGM and CGBS from 2011 to 2017, we had to deal with five different versions of the CGM and the CGBS. Consequently, the first task to do was to homogenize the measures and transform them into the 5.0 version. To do so, we employed the conversion table elaborated by the ECG advisors that were in charge of the development of the five versions of the model.

Table 1, below, depicts the dimensions (constructs/factors) and measures (items) that the ECG measurement theory proposes to manage and monitor sustainability and to measure the firms' relationships with their stakeholders in terms of social and environmental concerns.

**Table 1.** Dimensions and measurement scales of the Common Good Matrix (CGM) and Common Good Balance Sheet (CGBS).

| Dimension | Items | Measurement Scales |
|---|---|---|
| Suppliers A | A1. Human dignity in the supply chain.<br>A2. Solidarity and social justice in the supply chain.<br>A3. Environmental sustainability in the supply chain.<br>A4. Transparency and co-determination in the supply chain. | Absolute values (scores) |
| Owners, equity and financial service providers B | B1. Ethical position concerning financial resources.<br>B2. Social position concerning financial resources.<br>B3. Use of funds concerning the environment.<br>B4. Ownership and co-determination. | Absolute values (scores) |
| Employees C | C1. Human dignity in the workplace and the working environment.<br>C2. Self-determined working arrangements.<br>C3. Environmentally friendly behavior of staff.<br>C4. Co-determination and transparency within the organization. | Absolute values (scores) |
| Customers and business partners D | D1. Ethical customer relations.<br>D2. Cooperation and solidarity with other companies.<br>D3. Impact on the environment of the use and disposal of products and services.<br>D4. Customer participation and product transparency. | Absolute values (scores) |
| Social environment E | E1. Purpose of products and services and their effects on society.<br>E2. Contribution to the community.<br>E3. Reduction of environmental impact.<br>E4. Social co-determination and transparency. | Absolute values (scores) |

As no valid conclusions exist without valid measurement, our goal was to test the measurement theory proposed by the ECG model. Thus, we assessed whether the ECG model's theoretical specification of the factors matched the real observations using confirmatory factor analysis (CFA). According to Hair et al., CFA is an appropriate technique because it enables us to confirm or reject a preconceived measurement theory [39].

Consequently, following Hair et al. [31] and Ploum et al. [40], we proceeded to specify both the number of factors and observed variables according to the ECG model's measurement theory described in the previous sections. Thereafter, we assigned every observed variable or item to only one factor and ran the calculations by using IBM SPSS AMOS 23, we used the maximum likelihood robust extraction method as the estimator.

Moreover, Worthington and Whittaker [41] point to exploratory factor analysis (EFA) followed by CFA as being one of the most common approaches to scale development and validation. Therefore, we also took the EFA analysis that we had previously performed and published as a starting point [8].

Finally, we analyzed the results of CFA to assess their degree of generalizability. Specifically, in our research, the generalizability of the results would involve the empirical demonstration that the CGM and the CGBS are adequate (valid) tools to manage and report non-financial concerns.

## 4. Findings

Once we ran the software, the first step to proceed with CFA was to assess the goodness-of-fit statistics. Table 2 below, shows the goodness-of-fit statistics for the full model with 5 factors and 20 items.

**Table 2.** The CGM confirmatory factor analysis (CFA) goodness-of-fit statistics. Full set of 5 factors and 20 items.

| **Chi-Square Test** |
| --- |
| Chi-square = 1030.026 ($p$ = 0.000) |
| Degrees of freedom df = 170 |
| **Absolute Fit Measures** |
| Goodness of fit index (GFI) = 0.651 |
| Root mean square error of approximation (RMSEA) = 0.157 |
| 90% Confidence Interval for RMSEA = (0.148; 0.166) |
| Standardized root mean residual (SRMR) = 0.266 |
| Normed Chi-square = 6.060 |
| **Incremental Fit Measures** |
| Normed fit index (NFI) = 0.774 |
| Non-normed fit index (NNFI) = 0.780 |
| Comparative fit index (CFI) = 0.803 |
| Relative non-centrality fit index (RNI) = 0.803 |
| **Parsimony Fit Indices** |
| Parsimony normed fit index (PNFI) = 0.693 |
| Akaike (AIC) = 8221.429 |

As we can observe in Table 2, we did not face any identification problems as the degrees of freedom (df) value was above zero. Thus, the theoretical model had more unique covariance and variance terms than parameters to be estimated and, consequently, CFA will produce a stable solution [31].

Thereafter, we proceeded to assess the overall model goodness-of-fit. To do so, we relied on multiple fit indices [42]. Table 2 depicts absolute, incremental, and parsimony fit indices. Thus, to the Chi-square test, the p-value associated is below the recommended threshold of 0.05 [43]. Moreover, the Chi-square goodness-of-fit statistic did not indicate that the observed covariance matrix matches the estimated covariance matrix. However, as it is not advised to use this test alone, we examined other fit statistics.

Concerning other absolute fit indices, the goodness-of-fit index (GFI) was below the recommended threshold of 0.95 [44]. However, given the sensitivity of this index, some authors argue that it should not be employed [45]. For that reason, following Hooper et al. [42], we relied on the root mean square error of approximation (RMSEA), standardized root mean residual (SRMR), and normed Chi-square as absolute fit indices. As Table 2 shows, the RMSEA was above the guideline value of 0.08, as was the upper bond of the 90% RMSEA confidence interval; the SRMR was also above the 0.08 cutoff value and the normed Chi-square was above 5. Hence, the absolute fit measures did not provide us evidence to conclude that we were facing a model with acceptable goodness-of-fit.

Furthermore, following Hooper et al. [42], neither the incremental fit statistics nor the parsimony ones supported the existence of enough level of goodness-of-fit. Therefore, the empirical evidence was suggesting that the ECG measurement theory required some redefinition.

However, as the different goodness-of-fit indices provided were quite close to the cutoff values, it suggested that we were not so far and, thus, we proceeded to analyze where the possible causes of this lack of enough level of goodness-of-fit were. To do so, we followed the procedures described by Hooper et al. [42] and Hair et al. [31].

Then, we checked the standardized residuals and confirmed that none of them exceeded the ±4.00 benchmark that may indicate problems with the items affected. Instead, all the standardized residuals felt within the more conservative interval of ±2.5. From that, we concluded that the problem in reaching appropriate levels of goodness-of-fit was likely to be mostly caused by the factor definition and the association of the items according to the ECG measurement theory.

Thereafter, we analyzed the validity of the factors. Table 3, below, shows the standardized factor loadings, the average variance extracted, and the reliability statistics for the full set of 5 factors and 20 items.

**Table 3.** Standardized factor loadings, average variance extracted, and reliability estimates. Full set of 5 factors and 20 items.

| Factor | Indicator | Stand. Factor Loadings | AVE | Cronbach's $\alpha$ | Composite Reliability |
|---|---|---|---|---|---|
| SPLM | A1 | 0.997 * | 0.969 | 0.993 | 0.992 |
| | A2 | 0.996 * | | | |
| | A3 | 0.970 * | | | |
| | A4 | 0.974 * | | | |
| OFPM | B1 | 0.953 * | 0.897 | 0.976 | 0.972 |
| | B2 | 0.989 * | | | |
| | B3 | 0.883 * | | | |
| | B4 | 0.959 * | | | |
| EMPL | C1 | 0.328 * | 0.344 | 0.565 | 0.607 |
| | C2 | 0.916 * | | | |
| | C3 | 0.124 | | | |
| | C4 | 0.644 * | | | |
| CUST | D1 | 0.519 * | 0.330 | 0.631 | 0.644 |
| | D2 | 0.810 * | | | |
| | D3 | 0.355 * | | | |
| | D4 | 0.519 * | | | |
| SOCENV | E1 | 0.473 * | 0.288 | 0.567 | 0.579 |
| | E2 | 0.814 * | | | |
| | E3 | 0.232 * | | | |
| | E4 | 0.461 * | | | |

Note: * Significant at 0.05 level.

As we can observe in Table 3, the factors SPLM and OFPM corresponding to the dimensions A and B of the measurement theory described by the CGM showed average variance extracted (AVE) values above the threshold of 0.5 and reliability estimates above 0.7 [31]. Moreover, all the standardized loadings associated with those factors were above the 0.7 cutoff [46] and were statistically significant at the 0.05 level. Consequently, we concluded that dimensions A and B of the CGM were properly defined and the items correctly associated. Hence, we can affirm that SPLM and OFPM showed convergent validity.

On the contrary, the factors EMPL, CUST, and SOCENV corresponding to the dimensions C, D, and E of the CGM, showed AVE values below 0.5 and reliability statistics below 0.7. Moreover, we checked the loadings and found that some of the items showed weak and statistically non-significant loadings. Before advancing in the redefinition of these three factors, we tested whether they matched a formative design approximation by employing SmartPLS 3.2.7 software. Hence, we concluded that the constructs EMPL, CUST, and SOCENV did not match a formative design.

Thereafter, we redefined the constructs EMPL, CUST, and SOCENV taking a reflective design as a starting point. In this sense, according to Hooper et al. [42], these factors can be locally modified to improve the overall model fit based on removing those items showing $R^2$ below 0.2. For this reason, we checked the items $R^2$ and eliminated one by one those items that showed standardized loadings bellow 0.5 [31] and $R^2$ below 0.20. As a result, C1, C3, D3, E1, E3, and E4 were removed one by one from the model. After every iteration, we checked the goodness-of-fit statistics and construct reliability.

It is worth mentioning that the EFA [8] revealed important cross-loading problems concerning items C3, D3, and D3 that drove us to remove those items from the EFA analysis. In this sense, CFA confirmed EFA results. In the same way, the EFA solution included a factor with two items. However, according to Hair et al. [31], factors with fewer than three indicators should be avoided when applying CFA.

Therefore, taking the EFA results [8] into consideration, we proceeded to redefine the factors by merging dimensions C and E (EMPL and SOCENV). Thus, we respecified the ECG measurement theory by employing 4 factors (SPLM, OFPM, EMPL and SOC, and CUS) and 14 items. Figure 3 below shows the respecified model.

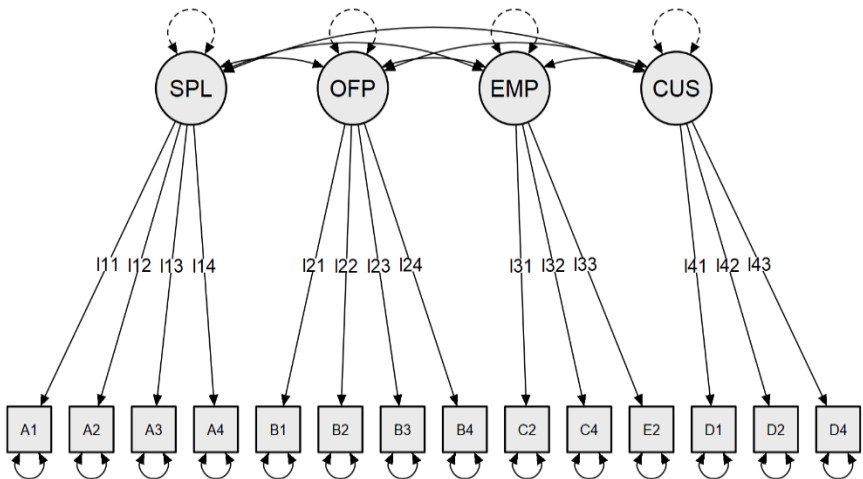

**Figure 3.** Model with 4 factors and 14 items.

Thereafter, we recalculated the results. Table 4 below depicts the goodness-of-fit statistics corresponding to the respecified model with 4 factors and 14 items.

As we can see in Table 4, we did not face any identification problems as the degrees of freedom (df) value was above zero. Therefore, the respecified model was overidentified and likely to produce a stable CFA solution.

Thereafter, we proceeded to assess the overall model goodness-of-fit. To do so, we checked multiple fit indices [42]. Table 4 provides measures of absolute, incremental, and parsimony fit indices. Concerning the Chi-square test, the p-value associated was below the recommended threshold of 0.05 [43]. Thus, the Chi-square goodness-of-fit statistic did not indicate that the observed covariance matrix matches the estimated covariance matrix. However, we examined other fit statistics.

Regarding other absolute fit indices, GFI was very close to the recommended threshold of 0.95 [44]. However, given the sensitivity of this index, some authors argue that it should not be employed [45]. For that reason, following Hooper et al. [42], we relied on RMSEA, SRMR, and normed Chi-square as absolute fit indices. As Table 2 shows, the RMSEA was below the guideline value of 0.08, as was the upper bond of the 90% RMSEA confidence interval. The SRMR was also below the 0.05 conservative cutoff value and the normed Chi-square was smaller than the conservative 2 cutoff value, hence confirming that the respecified model allowed to improve the absolute fit measures in comparison to the original, thus, providing evidence to conclude that we were facing a model with acceptable goodness-of-fit.

**Table 4.** The CGM CFA goodness-of-fit statistics. Set of 4 factors and 14 items.

| Chi-Square Test |
|:---:|
| Chi-square = 129.249 (*p* = 0.000) |
| Degrees of freedom df = 71 |
| **Absolute Fit Measures** |
| Goodness-of-fit index (GFI) = 0.943 |
| Root mean square error of approximation (RMSEA) = 0.019 |
| 90% Confidence Interval for RMSEA = (0.005; 0.034) |
| Standardized root mean residual (SRMR) = 0.047 |
| Normed Chi-square = 1.820 |
| **Incremental Fit Measures** |
| Normed fit index (NFI) = 0.929 |
| Non-normed fit index (NNFI) = 0.930 |
| Comparative fit index (CFI) = 0.964 |
| Relative non-centrality fit index (RNI) = 0.946 |
| **Parsimony Fit Indices** |
| Parsimony normed fit index (PNFI) = 0.725 |
| Akaike (AIC) = 5168.071 |

Moreover, following Hooper et al. [42], we checked the incremental fit statistics and the parsimony ones. Thus, all the incremental fit indices showed values above the 0.9 threshold and very close to the most conservative 0.95. As for the parsimony fit indices, Mulaik et al. [47] point out that parsimony fit indices above 0.5 while other goodness of fit indices achieve values over 0.90 can be interpreted as evidence of model parsimony. As shown in Table 4, the parsimony normed fit index (PNFI) for the respecified model was 0.725 whilst the absolute and incremental fit indices were above 0.9. Tables 2 and 4 also show the AKAIKE (AIC) statistic. The AIC is a non-normed statistic that does not fall into the interval 0–1, so it is more difficult to interpret. However, the model that produces the lowest AIC value is the most superior [48]. As we can observe in Tables 2 and 4, the AIC took a value of 8221.429 for the original ECG measurement model (5 factors and 20 items), whilst the respecified model (4 factors and 14 items) produced an AIC of 5168.071. Thus, we can conclude that the evidence supported the existence of an adequate level of goodness-of-fit in the respecified model.

Then, we assessed the validity of the four-factor solution produced by the respecified model. Table 5 shows the standardized factor loadings, the average variance extracted, and the reliability statistics for the respecified model.

As we can see, all the factors of the respecified model showed AVE above the 0.5 threshold and reliability estimates above 0.7. Moreover, all the factor loadings were above or close to the 0.7 cutoff and statistically significant at the 0.05 level, from which we concluded that the factors of the respecified model showed convergent validity.

Thereafter, following Hair et al. [31], we examined the discriminant validity of the respecified model. Table 6 depicts the correlation estimates among constructs, the AVE of every construct, and the constructs' squared correlations.

**Table 5.** Standardized factor loadings, average variance extracted, and reliability estimates. Set of 4 factors and 14 items.

| Factor | Indicator | Stand. Factor Loadings | AVE | Cronbach's $\alpha$ | Composite Reliability |
|---|---|---|---|---|---|
| SPLM | A1 | 0.997 * | 0.969 | 0.993 | 0.992 |
| | A2 | 0.996 * | | | |
| | A3 | 0.970 * | | | |
| | A4 | 0.974 * | | | |
| OFPM | B1 | 0.954 * | 0.897 | 0.976 | 0.972 |
| | B2 | 0.988 * | | | |
| | B3 | 0.884 * | | | |
| | B4 | 0.960 * | | | |
| EMPL and SOC | C2 | 0.909 * | 0.572 | 0.793 | 0.797 |
| | C4 | 0.654 * | | | |
| | E2 | 0.680 * | | | |
| CUST | D1 | 0.689 * | 0.512 | 0.704 | 0.715 |
| | D2 | 0.758 * | | | |
| | D4 | 0.697 * | | | |

Note: * Significant at 0.05 level.

**Table 6.** Discriminant validity. Set of 4 factors and 14 items.

| | SPL | OFP | EMPL and SOC | CUST |
|---|---|---|---|---|
| SPL | 0.969 | 0.165 | 0.040 | 0.149 |
| OFP | 0.406 * | 0.897 | 0.026 | 0.205 |
| EMPL&SOC | 0.201 * | 0.162 * | 0.572 | 0.271 |
| CUST | 0.386 * | 0.453 * | 0.521 * | 0.512 |

Note: * Significant at 0.05 level.

As we can see in Table 6, the AVE estimates for each factor were greater than the squared inter-construct correlations associated with that factor. Consequently, the factors included in the respecified model showed discriminant validity.

Finally, all the correlation estimates among constructs were statistically significant at the 0.05 level, so the factors were positively correlated one to another. Thus, we concluded that evidence in favor of the existence of nomological validity existed.

## 5. Discussion and Conclusions

The present work aimed to present the ECG measurement theory, which relies on the CGM and the CGBS as sustainability management and control tools, within the framework of corporate sustainability management tools and integrating reporting pointing to the model's ability to operationalize the SDGs in the business context.

Being the integration of the SDGs one of the main challenges in today's business reality, the ECG model arises as an alternative measurement theory to allow such integration into business practice. In this sense, some authors have recently linked the different cells and indicators of the CGM to the SDGs [37], thus providing evidence of face validity about the ECG measurement theory and its ability to integrate the SDGs into business management. However, concerning business practices, they did not provide empirical evidence to support their arguments. Thus, this paper tries to fill this gap by providing empirical evidence.

In this sense, as no valid conclusions can exist without valid measurement, our present work contributes to the advance of knowledge by conducting a CFA to assess how well the ECG measurement theory fits reality. It is based on a sample of 206 European firms that have implemented the model up to December 2017, so we consider it has the potential to produce some insights to scale the ECG measurement theory.

As a previous step to the CFA, we previously conducted an EFA to analyze the underlying structure [8]. One of the conclusions we got from EFA was the deletion of items C3, D3, and E3 due to cross-loadings concerns. CFA confirmed these results, as the inclusion of these three items in the model produced not reliable factors (AVE bellow 0.5 and reliability estimates bellow 0.7). To get to the reasons why this happened we should look at the definition of the item in the "Full Balance Sheet Workbook 5.0".

Thus, we find the indicator C3, related to environmentally friendly behavior of staff, that allocates the scores according to three criteria: i.e., the proportion of meals during the working hours that the staff gets from organic sources, the proportion of staff that commutes to work by car, public transport, bicycle, or on foot, and, finally, the take-up of environmentally friendly employee benefits. In regards to the first of the criteria, we found that it can also be reflecting somewhat affecting food suppliers (dimension A) or owners (dimension B) in the case of SMEs (most of the ECG firms population and sample are SMEs). Therefore, we advocate for the substitution of this criterion by another more clearly tied to staff environmental behavior. For example, the percentage of environmentally friendly processes carried out by staff [49,50] could be a good criterion to allocate the score of this item.

On the other hand, in the abovementioned workbook, item D3 is scored according to the impact on the environment of the use and disposal of products and services which overlaps issues related to the environmental management of the supply chain. That is the reason why the EFA [8] revealed the existence of cross-loadings concerning this item, and this item caused construct reliability concerns in CFA. Item E3 caused the problems following the same pattern as C3 and D3, as in the previously mentioned workbook it is scored according to criteria that are more related to supply chain operations than to business social environment (e.g., transport greenhouse gas emissions, fuel consumption, electricity consumption, paper consumption, chemicals, etc.).

Following, the item C1 (human dignity in the workplace and working environment), this item is scored in the workbook according to the degree of development of an employee-focused organizational culture, the degree of development of health promotion, occupational health and safety, and, finally, diversity and equal opportunities. Analyzing this item definition, we consider that, maybe, health-related concerns could be low correlated with organizational culture and diversity and equal opportunities. Therefore, putting together these criteria to score the item may cause some problems of face validity, and thereafter it may cause problems of convergent validity.

Moreover, according to the workbook's definition, item E1 measures issues related to the purpose of products and services and their impact on society. To do so, the score is allocated following these criteria: product and services should cover basic needs and contribute to a good life, the social impact of products and services, and finally, unethical and unfit products and services. Once again, in our opinion the abovementioned criteria may cause problems of face validity as some of the criteria employed are related to other stakeholder groups considered in the model, i.e., we see the criteria product and services should cover basic needs and contribute to a good life, and unethical and unfit products and services, more directly tied to customers than to the social environment.

In regards to the item E4 (social co-determination and transparency), the workbook allocates its score according to the following criteria: the degree of transparency, especially about the introduction of new production processes which involve hazardous substances or significant environmental impact, social participation through stakeholder's share of co-decision making, and lack of transparency and willful information. In this case, we find that it was also the overlap of underlying concepts which brought to a lack of face validity to the item because the criteria employed to allocate the score had to do with other stakeholders.

From all that has been pointed out above, we concluded that those items that we removed from the original model suffered from a lack of face validity and, consequently, their inclusion in the measurement theory was the source of the factors' lack of convergent validity and this additionally caused the poor level of goodness-of-fit when we applied CFA to the original ECG measurement theory.

Moreover, the merging of dimensions related to employees (C), and social environment (E) into a combined dimension renamed as "employees and social environment" was made based on the score

allocation criteria concerning item E2 given in the above-mentioned workbook [7]. Specifically, item E2 was scored taking the net tax ratio as a base which, in turn, depends on payroll tax and social security contributions paid by employers, income tax, and social security contributions paid by employees. Thus, we stated that the score allocation of item E2 was based on criteria related to employees. This fact, together with the EFA results, made us decide to merge both dimensions including the items with standardized factor loadings over 0.5, and $R^2$ over 0.2 i.e., C2, C4, and E2. This way we ensured the construct face validity.

In short, the present research has allowed us to assess the ECG measurement theory and identify the items that were causing problems to consider such measurement theory as valid and reliable to manage and monitor sustainability in the business context. Thereafter, we have respecified the measurement theory to reach a valid and reliable solution so that the modified model can still be employed for the purpose for which it was conceived. Future research should redefine the items that have been removed from the model and retest the measurement theory with the redefined items.

However, it is worth mentioning that two of the factors included in the original model (SPLM and OFP) were fully validated by employing CFA. This means that the ECG measurement theory provided effective measurement scales to manage and monitor the sustainable management of the supply chain and, also, of the business financials allowing the integration of SDGs. Consequently, our work contributes to the existing research body at the intersection of business and SDGs by validating some measurement scales aimed at the operationalization of the SDGs in the business practice. As literature has pointed to the lack of understanding of how to operationalize SDGs in the business context as one of the existing research gaps, the present work makes a significant contribution in such field research [51–53].

**Author Contributions:** Conceptualization, A.T.E. and V.C.; methodology, V.C.; software, A.T.E. and V.C.; validation, A.T.E. and V.C.; formal analysis, A.T.E. and V.C.; investigation, A.T.E. and V.C.; resources, V.C.; data curation, A.T.E.; writing—original draft preparation, A.T.E. and V.C.; writing—review and editing, A.T.E. and V.C.; visualization, A.T.E. and V.C.; supervision, V.C.; project administration, A.T.E. and V.C.; funding acquisition, V.C. All authors have read and agreed to the published version of the manuscript.

**Funding:** This research was funded by Humanistic Management Practices gGmbH.

**Conflicts of Interest:** The authors declare no conflict of interest.

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
