# Peer review of "Assessing the Economy for the Common Good Measurement Theory Ability to Integrate the SDGs into MSMEs"

_sustainability, doi:10.3390/su122410305_

Round 1

Reviewer 1 Report

ABSTRACT

The abstract is incomplete. It is necessary to refer the methodology in a more explicit way and identify the main conclusion and contribution of the research.

This abstract is very synthetic and the reader can't understand well what is intended.

INTRODUCTION

Very well elaborated, with relevant references. The way that authors present this introduction allows an immediate understanding of what is to be investigated.

Note: Please note the two abbreviations IR and SDGs that are not defined the first time they are mentioned in the text. You should correct this situation

THEORICAL FRAMEWORK

Very good literature review with presentation of relevant studies. 

The presentation of the structural model proposed at the end of the literature review is appealing and I agree with its introduction at the end, although this situation is not very common. Usually the proposed model is in its own section.

METHODOLOGY

Well explained, cites authors who used CFA. Nothing to point out.

FINDINGS

1- My biggest preoccupations with this article are in this section. The authors propose a model with 5 factors, each factor with specific questions that measure each dimension.

2-The authors test the measurement model and verify that some dimensions do not present reliability indices such as (CFI,GFI,AVE,RMSEA, etc.) capable of validating the model with 5 factors.

3-Procedes the removal of factors with loadings below 0.5. So far, everything is correct and well done.

4-How the model did not have enough indices to be validated the authors decided to maintain (and very well) the SPL and OFP dimensions and how the other dimensions had indices, for example of AVE, below what is acceptable they decided to add the EMPL&SOC dimensions. In my opinion this is a serious research error that needs to be corrected and I explain why:

a) The EMP dimension measures Employees 

C1. Human dignity in the workplace and the working environment.

C2. Self-determined working arrangements.

C3. Environmentally friendly behavior of staff.

C4. Co-determination and transparency within the organization.

b) The SOC dimension measures the Social environment

E1. Purpose of products and services and their effects on society.

E2. Contribution to the community.

E3. Reduction of environmental impact.

E4. Social co-determination and transparency.

IMPORTANT WARNING: The authors decide to join EMPL&SOC, composed by the variables C2, C4 and E2 where E2 has nothing to do with C2 and C4. Now, they are putting together variables of dimensions that measure completely different realities. This is not acceptable. If the dimension has no statistical validity in the model there is only one solution (it has to be removed). We cannot put together variables that measure different things and give it a joint name. This is not acceptable and no author who has used CFA has done this.

When the authors verified that there were dimensions that did not present acceptable results they should remove them and test the model again.

In fact, the authors did all right with the remaining dimensions: they removed the variables with loadings below 0.5 and re-tested. But with the above dimensions they made a methodological research error that must be corrected for the article to be accepted.

In short, the dimensions cannot be tested together because they address completely different realities.

If the dimensions are not valid, they must be removed and a final/scale model with robust dimensions must be proposed.

I'm sorry, but from here on the discussion is completely out of sense because there is a previous serious mistake.

I ask the authors to correct this error and show me how the fine model looks and its statistical validity.

Good luck with this article

Reviewer 2 Report

In my view this is a well crafted paper.

The review of literature sets up the rationale for the study well and the problem framed in this rationale is well articulated. There is a good deal of literature that highlights the importance of treating SMEs and MSMEs as heterogeneous, so perhaps this study might pay more attention to the challenges of creating standard items/scales that apply in such diverse contexts.

The sample is solid and the description of methodology clear.

The presentation is good throughout, although there are a few places where language can be improved. A thorough proof read will certainly address this.

However, the conclusion could be improved by more clearly setting out the implications for the existing CGM, CGBS and ECG measurement theory. This is more about clarity of expression than the content itself. It might also discuss how the findings can be applied to the useful tools that are already in existence.

Reviewer 3 Report

The article is a high-quality analysis of Common Good Measurement Theory implementation In Europe (204 companies). Having in mind the Green Deal Strategy in the Circular Economy context CGMT has crucial importance. Thus the article presents a very high interest to the readers and for economic agents.

The theoretical framework reveals the current context and presents Common Good Matrix used by the economic agents (especially private companies) to achieve Sustainable Development Goals.

The authors designed a model based on 5 factors and 20 items, extracted from Common Good Matrix, and made a factor analysis to measure the importance of each factor. In the second step of the analysis the eliminated C1, C3, D3, E1, E3, and E4 items as being irrelevant, or probably improper scaled/ evaluated. The second model based on 4 factors (SPLM, OFPM, EMPL&SOC and, SOCENV) and 14 items is very coherent as all the statistics proved (Chi-square test,
Goodness of fit index, RMSEA, Normed fit index, Parsimony normed fit index, Akaike).

Another very strong part of the article is the conclusions, that come with coherent explanations in accordance with statistical results. The authors explained how they manage to substitute removed criteria with other more appropriate or why other criteria may cause problems of face validity.

Finally, the authors designed a coherent,  valid and reliable model that might be used with success in Common Good Measurement Theory implementation by companies to achieve the Green Deals Goals, eliminating items that were causing problems to consider such measurement theory.

Please pay attention to the next phrase (Line305):" The average score obtained by the firms included in the sample was 497, the median was 498; which means that, according to the rating employed by the CGBS, most of them fall into the “experienced” level (between 301 and 600 points). " I should be detailed and rearranged in 2 phrases because it is difficult to observe the dependency between them. The mean and median (and module) talks about kurtosis and skewness, but not about categories of firms (experienced, exemplary, beginners, etc)

I also would recommend the authors to read other scientifically researched published in the last 3 years.  

Reviewer 4 Report

The paper is well organised. The introduction is wide and well locates the topic of the paper in the scientific international research context. The theoretical framework gives a clear support to the proposed research. The paper provides an additional contribute to the enhancement of the knowledge in the field of Common Good.

Round 2

Reviewer 1 Report

Dear authors,

First of all, I would like to clarify that I never questioned the quality of your work. The research, from the point of view of the organization and content is excellent and is very interesting. My only doubt concerns the methodological part from the theoretical point of view with implications for the statistical analyzes carried out through the EFA and CFA analysis .

Although I still have doubts about the validity of the combination of factors, I believe that your explanation and clarification in the article are valid. In this sense, believing that theoretically this junction can be made, transforming two different factors (which in my opinion should be separated) into just one, I accept that they do so, but I leave the following doubt: Can a single variable have explanatory capacity to maintain both factors now together? Wouldn't it be better to drop this variable and consequently this dimension? I just leave this question for future reflection, in future works that use this scale.

In conclusion: an excellent job with only one situation that, despite your extensive statistical explanation, confuses me.

 However, congratulations from your investigation.

All the best in future 

Best Regards 

Author Response

Dear reviewer,

we see your review as an opportunity to improve those parts of our work that, maybe, were not clear enough. In this sense, we'd like to thank you for pointing out these issues, and for contributing to improving our draft manuscript. 

As for the matter you comment on in this second round, eliminating one of the items (E2) would involve eliminating an entire construct "EMPL&SOCENV" because in CFA constructs must have associated, at least, 3 items (Hair et. al., 2018)

Therefore, eliminating E2 involved eliminating C2 and C4. At the moment we were working on our research we tried this option. However, this option didn't provide an appropriate level of GOF concerning the entire model. This happened because, this way, lots of information included in the original model was lost in the process. As our purpose was to statistically validate the ECG theory in a way that we can reach a model that could still be applied by practitioners, we looked for a balance between retaining the maximum number of items while reaching an appropriate level of GOF and, ensuring face validity in the constructs included in the model.

Finding this equilibrium took us time as we deeply analyzed the items' definitions provided in the workbooks to try to give a proper ground to our work.

All the best for the future.

Sincerely,

The authors.